



# Wave energy attenuation in fields of colliding ice floes. Part A: Discrete-element modelling of dissipation due to ice–water drag

Agnieszka Herman[1], Sukun Cheng[2], and Hayley H. Shen[3]

[1]Institute of Oceanography, University of Gdansk, Poland
[2]Nansen Environmental and Remote Sensing Center, Bergen, Norway
[3]Department of Civil and Environmental Engineering, Clarkson University, Potsdam, NY, USA

**Correspondence:** Agnieszka Herman (oceagah@ug.edu.pl)

**Abstract.** The energy of water waves propagating through sea ice is attenuated due to nondissipative (scattering) and dissipative processes. The nature of those processes and their contribution to attenuation depends on wave characteristics and ice properties, and is usually difficult (or impossible) to determine from limited observations available. Therefore, many aspects of relevant dissipation mechanisms remain poorly understood. In this work, a discrete-element model (DEM) is used to study one
of those mechanisms: dissipation due to ice–water drag. The model consists of two coupled parts, a DEM simulating the surge motion and collisions of ice floes driven by waves, and a wave module solving the wave energy transport equation with source terms computed based on phase-averaged DEM results. The wave energy attenuation is analyzed analytically for a limiting case of a compact, horizontally confined ice cover. It is shown that the usage of a quadratic drag law leads to nonexponential attenuation of wave amplitude $a$ with distance $x$, of a form $a(x) = 1/(\alpha x + 1/a_0)$, with the attenuation rate $\alpha$ linearly
proportional to the drag coefficient. The dependence of $\alpha$ on wave frequency $\omega$ varies with the dispersion relation used: for the open-water ('ow') dispersion relation, $\alpha \sim \omega^4$; for mass-loading dispersion relation, suitable for ice covers composed of small floes, the increase of $\alpha$ with $\omega$ is much faster than in the 'ow' case, leading to very fast elimination of high-frequency components from the wave energy spectrum; for elastic-plate dispersion relation, suitable for large floes or continuous ice, $\alpha \sim \omega^m$ within the high-frequency tail, with $m$ close to 2.0–2.5, i.e., dissipation is much slower than in the 'ow' case. The
coupled DEM–wave model predicts the existence of two zones: a relatively narrow area of very strong attenuation close to the ice edge, with energetic floe collisions and therefore high instantaneous ice–water velocities; and an inner zone where ice floes are in (semi)permanent contact with each other, with attenuation rates close to those analyzed theoretically. Dissipation in the collisional zone increases with increasing restitution coefficient of the ice and with decreasing floe size. In effect, two factors contribute to strong attenuation in fields of small ice floes: lower wave energy propagation speeds and higher relative ice–water
velocities due to larger accelerations of floes with smaller mass and more collisions per unit surface area.

## 1 Introduction

As ocean waves propagate through sea ice, they undergo attenuation due to both non-dissipative and dissipative processes. Whereas attenuation due to non-dissipative scattering has been extensively studied and can be regarded as well understood (see, e.g., Squire, 2007; Kohout and Meylan, 2008; Montiel et al., 2016; Montiel and Squire, 2017, and references there), many



aspects of dissipative processes accompanying wave propagation in sea ice remain relatively unexplored – even though observations leave little doubt that they play an important role in wave propagation in the marginal ice zone (MIZ). Understanding those processes and parameterizing their effects in models is important not only for reproducing/predicting local conditions in the MIZ, but also at much larger scales (see, e.g., the recent model sensitivity study by Bateson et al., 2019, who showed that

the simulated sea ice extent and volume in the Arctic is very sensitive to the wave attenuation rates).

Depending on wave forcing and sea ice properties, the relative importance of individual dissipative processes varies; similarly, the relative contribution of scattering and dissipation to the overall wave attenuation is strongly dependent on wave and ice conditions. In general, attenuation due to scattering at floes' edges tends to dominate at relatively low ice concentration and, crucially, when the floe sizes are comparable with wavelengths (Kohout, 2008; Kohout and Meylan, 2008). In compact ice in

the inner parts of the MIZ, scattering is induced at cracks and locations of rapid changes of ice thickness (e.g., pressure ridges; Bennetts and Squire, 2012). Processes leading to the dissipation of wave energy take place within the ice itself as well as in the underlying water layer and include viscous deformation of the ice, overwash, vortex shedding and turbulence generation, friction between ice floes and between ice and water (form and skin drag), inelastic floe–floe collisions, breaking and rafting of floes, and many more. Although, in some situations, the observed characteristics of waves in ice (change of wave height with

distance, directional distribution of wave energy, etc.) can be satisfactorily explained by non-dissipative scattering, taking into account dissipation is usually necessary to obtain agreement between observations and models, especially in grease/pancake ice or small floes (e.g., Liu and Mollo-Christensen, 1988; Rogers et al., 2016; Squire and Montiel, 2016; De Santi et al., 2018; Sutherland and Dumont, 2018; Sutherland et al., 2018a).

Considering the multitude of processes contributing to wave energy attenuation, it is not surprising that the observed attenu-

ation rates span a few orders of magnitude (e.g., Rogers et al., 2016; Stopa et al., 2018b). Although the available observational data on wave energy attenuation in sea ice has been growing since the 1980s and includes measurements performed with buoys (e.g., Wadhams et al., 1988; Liu et al., 1991; Cheng et al., 2017a; Montiel et al., 2018), airborne SAR and scanning lidar (Liu et al., 1991; Sutherland et al., 2018b), underwater ADCP (Hayes et al., 2007), and satellite SAR (Ardhuin et al., 2015; Stopa et al., 2018b, a), the interpretation of the observed attenuation rates is extremely difficult, as it would require simultaneous

measurements of several wave and ice characteristics over large distances. For example, although SAR data provide information on the variability of wave height and direction over large spatial domains, the lack of accompanying spatial distribution of ice properties (thickness, floe size, elastic modulus, etc.) makes inferences regarding possible causes of that variability very difficult.

Among the most crucial characteristics of wave energy attenuation in sea ice are: the functional dependence of wave ampli-

tude $a$ on distance travelled, $a(x)$, and the functional form describing the dependence of the respective attenuation coefficient $\alpha$ on wave frequency, $\alpha(\omega)$.

In most studies, exponential attenuation is assumed, and no alternative forms of $a(x)$ are considered. This is in many cases a well motivated choice. The exponential model does successfully represent observations in several of the studies cited above. In many cases, large scatter in observational data and/or limited number of measurement locations make the usage of more

complicated models unjustified. Also, several attenuation processes, including scattering, do lead to exponential attenuation.





On the other hand, however, some observations can hardly be represented by an exponential curve with a single attenuation rate over long distances (see, e.g., Stopa et al., 2018a; Ardhuin et al., 2018, who fitted separate exponential curves to data from locations close to the ice edge and from those further into the ice). Several potentially relevant mechanisms leading to non-exponential attenuation have been identified in theoretical studies, including ice–water friction relevant for this work (Shen

and Squire, 1998; Kohout et al., 2011). In a recent review paper, Squire (2018) discusses a more general formula:

$$\mathrm{d}a/\mathrm{d}x = -\alpha a^n, \tag{1}$$

which produces exponential attenuation, $a = a_0 \exp(-\alpha x)$, for $n = 1$ and has a solution $a^{1-n} = a_0^{1-n} - (1-n)\alpha x$ for $n \neq 1$.

Another problem with some models predicting exponential attenuation is related to the second of the two attenuation characteristics mentioned above – they produce $\alpha(\omega)$ that do not agree with observations. Most observational data suggest a power

law dependence, $\alpha \sim \omega^m$, with an exponent $m$ in the range 2–4, which is much lower than predicted by several widely used sea ice models (Meylan et al., 2018). The importance of the $\alpha(\omega)$ behavior in interpreting the observed wave energy attenuation in sea ice has been analyzed, e.g., by Meylan et al. (2014) and Li et al. (2015).

In this work – described below and in the companion paper (Herman *et al.* 2019, *The Cryosphere Discuss.*, paper ID tc-2019-130), referred to further as Part B – we combine discrete-element modelling (DEM) and laboratory experiments to study

selected aspects of attenuation and dissipation of wave energy in fragmented sea ice. The DEM model is that of Herman (2016), with wave forcing formulated by Herman (2018). It simulates wave-induced surge motion of ice floes of arbitrary sizes, and is used here with several necessary modifications described further. The laboratory experiments, analyzed in Part B, were performed as part of the international project Loads on Structure and Waves in Ice (LS-WICE; see Cheng et al., 2017b) and include tests related to propagation and attenuation of regular waves through fields of densely packed ice floes of equal sizes.

The present study is to a large extent motivated by the results of Herman (2018), who studied wave-induced floe collisions in highly idealized conditions (regular waves, constant wave amplitude, periodic domain boundaries, etc.), but with forcing formulated by integrating dynamic pressure and stress acting on each floe over the floe's surface area – as opposed to earlier similar models, in which forcing was specified for the center of mass of each floe. It was shown that this seemingly minor difference enabled the model to reproduce the amplitudes of the surge motion of floes with sizes comparable with wavelength.

Obviously, as the floe size increases, the floes are not able to follow the oscillating motion of the surrounding water, which leads to high ice–water velocity differences, which in turn might lead to substantial stress at the ice–water interface, depending on the values of the skin and form drag coefficients. Most importantly from the point of view of the present study, Herman (2018) demonstrated that the presence of collisions strongly enhances ice–water drag through two mechanisms, the relative importance of which depends on ice concentration, ice mechanical properties, floe size, and wave characteristics. One mech-

anism dominates in ice composed of small floes with large restitution coefficient, i.e., in situations with energetic collisions that lead to high post-collisional velocities of the floes. The second mechanism is particularly effective in ice composed of large, densely packed floes, when neighboring floes stay in contact over prolonged periods of time, so that the contact forces are non-zero over a substantial fraction of the wave period. Although the temporal variability of ice–water velocities in those two extremes is very different – with very high, but short-lived peaks in the first case, and less extreme, more uniformly time-



distributed values in the second case – the overall, phase-averaged effect is comparable in both cases and leads to significantly enhanced drag forces. This observation led Herman (2018) to speculate that this mechanism, based on an interplay of floe–floe collisions and ice–water drag effects, might contribute to dissipation of wave energy. To elucidate this idea in more detail is one of the purposes of the present study. To this end, we couple the DEM sea ice model with a simple wave attenuation model

(in a manner similar to that in Shen and Squire, 1998), and study the dynamics of ice floes and wave energy dissipation for a wide range of combinations of parameters. The overall setup of the model corresponds to that of the laboratory experiment mentioned above. We use the results of numerical simulations and, in Part B, laboratory observations to investigate several aspects of wave energy dissipation in sea ice. As already mentioned, one of the major specific goals is to analyze details of dissipation due to ice–water drag and collisions of ice floes. Another goal, on which we focus in Part B, is to illustrate how

even in seemingly very simple settings wave propagation and attenuation in sea ice is shaped by several interrelated processes, impossible to isolate from each other – and how several very different model configurations can be fitted to satisfactorily reproduce the observed wave attenuation rates, making identification of processes actually responsible for dissipation a formidable task.

After formulating the assumptions and equations of the sea ice and wave model in the next section, we begin our study

with a theoretical analysis of energy dissipation induced by ice–water drag in a special, limiting case of waves propagating through horizontally confined ice (i.e., with zero horizontal velocity). We show that the attenuation equation in this case can be solved analytically, and that this model configuration leads to non-exponential attenuation of the form (25), with $\alpha(\omega)$ strongly dependent on the assumed dispersion relation. This result is particularly interesting in view of the results of Cheng et al. (2018), who showed that the dispersion relation is strongly affected by floe size, with the wavenumber $k$ increasing with decreasing

floe length. The DEM results are presented in section 4. We begin with an analysis of the model sensitivity to changes of parameters, including ice concentration, restitution coefficient, drag coefficient, and floe size; we also discuss in detail a typical shape of the attenuation curve, which in many cases reflects the existence of two clearly distinct regions – a narrow zone close to the ice edge with strong collisions and very strong dissipation, and an inner zone with densely packed floes staying in semi-permanent contact with their neighbors and with slower attenuation, close to the theoretical solution mentioned above.

We discuss the modelling results in the context of recent research on wave attenuation in sea ice in section 5.

## 2 Model description

As already mentioned in the introduction, the model used here consists of two coupled parts, a sea ice module and a wave module. The sea ice part is based on the DEM model by Herman (2018), with modifications described below. The coupled model is one-dimensional and considers only the horizontal (surge and drift) motion of ice floes.




## 2.1 Definitions and assumptions

We consider linear, unidirectional, progressive waves with period $T$, propagating in the positive $x$-direction:

$$\eta = a(x)\cos\theta, \tag{2}$$

$$u_w = a(x)\omega\frac{\cosh[k(z+h)]}{\sinh[kh]}\cos\theta = u_{w,0}(x,t)\frac{\cosh[k(z+h)]}{\cosh[kh]}, \tag{3}$$

$$w_w = a(x)\omega\frac{\sinh[k(z+h)]}{\sinh[kh]}\sin\theta = w_{w,0}(x,t)\frac{\sinh[k(z+h)]}{\sinh[kh]}, \tag{4}$$

$$\theta = kx - \omega t, \tag{5}$$

where $\eta$ denotes the instantaneous water surface elevation relative to still water level at $z=0$, $(u_w, w_w)$ are the components of the water velocity vector in the $xz$ plane, $(u_{w,0}, w_{w,0})$ are velocity components at $z=0$, $t$ denotes time, $a$ is the $x$-dependent wave amplitude, $k=2\pi/L_w$ the wave number, $L_w$ denotes wavelength, $\omega=2\pi/T=2\pi f$ the wave angular frequency, and $\theta$ denotes phase.

The angular frequency and the wavenumber are related by the following dispersion relation (see, e.g., Fox and Squire, 1990; Collins et al., 2017):

$$\omega^2\left(1+\beta_1 k\tanh[kh]\right) = (g+\beta_2 k^4)k\tanh[kh], \quad \text{with} \quad \beta_1 = \frac{\rho_i}{\rho_w}h_i \quad \text{and} \quad \beta_2 = \frac{Eh_i^3}{12\rho_w(1-\nu^2)}, \tag{6}$$

where $g$ denotes acceleration due to gravity, $\rho_w$ is the water density, $\rho_i$ ice density, $h_i$ ice thickness, $E$ elastic modulus, and $\nu$ the Poisson's ratio. The corresponding group velocity $c_g \equiv d\omega/dk$ is given by:

$$c_g = \frac{\omega}{2k}\left[\left(1-\frac{\beta_1\omega^2}{g+\beta_2 k^4}\right)\left(1+\frac{2kh}{\sinh[2kh]}\right)+\frac{4\beta_2 k^4}{g+\beta_2 k^4}\right]. \tag{7}$$

In its full form, when $\beta_1$ (the inertial coefficient) and $\beta_2$ (the flexural rigidity) are different from zero, equations (6) and (7) describe waves propagating in water covered with an elastic plate. If $E=0$ and thus $\beta_2=0$, (6) and (7) reduce to the mass-loading model, which further reduces to open water waves when $\beta_1=0$ (i.e., $h_i=0$ and no ice is present). The elastic plate and mass loading models will be used in this study as two limiting cases, one suitable for situations with very large floes that undergo flexural motion (note that although the DEM disregards the vertical deflection of the floes, its influence on wave length and group velocity are taken into account), and the second one suitable for very small and non-interacting floes behaving as rigid floating objects. Although, in general, the open water case is not relevant for ice-covered seas, it is very useful as reference (importantly also, the wavenumbers observed in several tests of the experiment discussed in Part B were very close to open water values). In the rest of the paper, indices 'ep', 'ml' and 'ow' will be used to designate wavenumber and group velocity from a particular model ($k_{ep}$, $c_{g,ep}$, etc.); symbols without index will be used in a more general context, when no particular model is assumed.

It must be noted that in the case of small-amplitude, irrotational water waves propagating under multiple elastic, non-colliding plates floating on the surface, the velocity potential – and thus the velocity components – can be expressed, for each plate, as a sum of transmitted and reflected waves, each in turn consisting of traveling, damped traveling and evanescent modes



(see, e.g., Kohout and Meylan, 2008). Using equations (2)–(5) with dispersion relation (6) amounts to taking into account only the transmitted ('zeroth') component and omitting the remaining ones. In other words, it amounts to disregarding all scattering effects. The consequences of this simplification will be discussed in the last section and, in the context of the experimental data, in Part B.

As already mentioned, the model is one-dimensional, i.e., the ice floes are placed along the $x$ axis and indexed in such a way that the $i$-th floe neighbors the $(i-1)$-th and the $(i+1)$-th floes in the negative (upwave) and positive (downwave) $x$ direction, respectively. The floes are cuboid rigid bodies and their total number is $N_f$. Although the DEM allows for specifying different properties for each discrete element, in this study all floes have identical density $\rho_i$, thickness $h_i$, length $L_x = 2r_i$, width $L_y$, and mass $m_i = 2r_i L_y h_i \rho_i$. The thickness of the submerged part of each floe equals $h_i \rho_i / \rho_w$, i.e., Archimedean balance is

assumed. Apart from the elastic modulus $E$ and Poisson's ratio $\nu$, the ice is characterized by its restitution coefficient $\varepsilon$. As said, the model describes the horizontal (surge) motion of ice floes. Thus, the relevant time-dependent variables for each floe are the horizontal position of its center of mass, $x_i$, and its horizontal translational velocity, $u_i$.

## 2.2   Discrete element sea ice model

As in Herman (2018), the model solves the linear-momentum equations for each ice floe, with four types of forces:

$$m_i \frac{\mathrm{d}u_i}{\mathrm{d}t} = F_{\mathrm{w},i} + F_{\mathrm{v},i} + F_{\mathrm{d},i} + F_{\mathrm{c},i}, \qquad i = 1, \ldots, N_f, \tag{8}$$

where $F_{\mathrm{w},i}$ denotes the wave-induced force (Froude–Krylov force), $F_{\mathrm{v},i}$ – the virtual (or added) mass force, $F_{\mathrm{d},i}$ – the drag force, and $F_{\mathrm{c},i}$ – the sum of contact forces from all collision/contact partners of floe $i$. A detailed discussion on formulation of these forces can be found in Herman (2018) and will not be repeated here. The only difference with respect to the previous study concerns the computation of the drag force $F_{\mathrm{d},i}$. Due to reasons of computational efficiency, Herman (2018) proposed

an approximate formula to avoid numerically integrating the local ice–water stress over the bottom surface of each floe at each time step. In the present study, a very similar formula is required for computation of both $F_{\mathrm{d},i}$ and the energy dissipation term in the wave-energy equation (see further section 2.3). Thus, for the sake of consistency between the different model parts, the integrals in both cases are computed numerically, with the same spatial resolution.

## 2.3   Wave energy attenuation

As marked explicitly in (2)–(4), the wave amplitude in the present model varies in space, $a = a(x)$. It is assumed that the amplitude at the ice edge (corresponding to the position of the left side of the first floe, $x = x_1 - r_1$) is known and equals $a_0$. At the remaining locations, $a$ is determined from the energy conservation equation:

$$\frac{\mathrm{d}}{\mathrm{d}x}(c_g E_w) = \sum_m S_{\mathrm{dis},m}, \tag{9}$$

where the wave energy $E_w$, in J/m$^2$, is given as:

$$E_w = \frac{1}{2}\rho_w g a^2 \tag{10}$$



and the source terms on the right-hand-side of (9) represent phase-averaged dissipation rate per unit area of an ice floe, expressed in W/m$^2$. In this work, two source terms are considered. The first one ($S_{\mathrm{sd}}$), of particular interest in this study, describes energy dissipation due to skin drag at the ice–water interface. The second one ($S_{\mathrm{ow}}$), included for the purpose of the laboratory case study analyzed in Part B, describes energy losses due to overwash. Thus, from (9) and (10):

$$\frac{1}{2}\rho_w g c_g \frac{\mathrm{d}(a^2)}{\mathrm{d}x} = S_{\mathrm{sd}} + S_{\mathrm{ow}} \tag{11}$$

so that, assuming constant dissipation over a certain (small) distance $x$, the amplitude at $x_0 + x$ can be computed from the amplitude at $x_0$ as:

$$a(x_0 + x) = \left[ a^2(x_0) + \frac{2(S_{\mathrm{sd}} + S_{\mathrm{ow}})}{\rho_w g c_g} x \right]^{1/2}. \tag{12}$$

Note that, different than in the study by Shen and Squire (1998), $E_w$ denotes the energy of the waves, not the energy of the whole water and ice system. This justifies the usage of the group velocity $c_g$ in (9) as the energy-transport velocity and of the formula (10) relating $E_w$ to the wave amplitude $a$. Crucially, this is the reason why no source term is present in (11) explicitly describing dissipation due to inelastic collisions. The inelastic collisions influence the wave propagation through their influence on ice velocity, which in turn modifies the ice–water drag. This makes the model different from that of Shen and Squire (1998).

### 2.3.1 Dissipation due to ice–water drag

For an individual ice floe with bottom surface area $A_{\mathrm{bot}}$, $S_{\mathrm{sd}}$ can be obtained from (see, e.g., Shen and Squire, 1998):

$$S_{\mathrm{sd}} = -\frac{1}{n_T T}\frac{1}{A_{\mathrm{bot}}} \int\limits_{t_0}^{t_0 + n_T T} \int\limits_{A_{\mathrm{bot}}} \tau_w u_{\mathrm{rel}} \mathrm{d}s \mathrm{d}t, \tag{13}$$

where $n_T$ is an integer (i.e., the averaging is performed over a multiple of the wave period $T$), $u_{\mathrm{rel}}$ denotes the module of the local, instantaneous ice–water velocity difference:

$$u_{\mathrm{rel}}(x,t) = |u_i(t) - u_{w,0}(x,t)|, \tag{14}$$

and $\tau_w$ denotes the module of the local ice–water stress. In this study we use the quadratic drag law:

$$\tau_w = \rho_w C_{\mathrm{sd}} u_{\mathrm{rel}}^2 \tag{15}$$

and assume that the drag coefficient $C_{\mathrm{sd}}$ is constant.

### 2.3.2 Dissipation due to overwash

We use a very unsophisticated approximation of overwash effects, the development of which was motivated by the observation that strong overwash occurred in laboratory tests analyzed in Part B. The algorithm described here should be treated as a framework for future parameterizations rather than as an ultimate solution.





Following Skene et al. (2018), the energy flux (in N/s) due to overwash, $\dot{E}_{\text{ow}}$, consisting of the kinetic and potential energy parts, can be expressed in terms of the average overwash velocity $u_{\text{ow}}$ and depth $h_{\text{ow}}$:

$$\dot{E}_{\text{ow}} = u_{\text{ow}} h_{\text{ow}} \left( \frac{1}{2} \rho_w u_{\text{ow}}^2 + \rho_w g h_{\text{ow}} \right). \tag{16}$$

The results of Skene et al. (2015, 2018) justify an assumption that overwash behaves as a shallow water wave propagating over the upper surface of the ice, so that $u_{\text{ow}} = (g h_{\text{ow}})^{1/2}$ and:

$$\dot{E}_{\text{ow}} = \frac{3}{2} \rho_w g^{3/2} h_{\text{ow}}^{5/2}. \tag{17}$$

Estimating $h_{\text{ow}}$ is the most problematic part of the algorithm. It involves two issues: first, a criterion for the overwash to occur (i.e., the conditions for $h_{\text{ow}} > 0$), and second, how $h_{\text{ow}}$ depends on wave and ice conditions. In this study, one of the simplest expressions possible is adopted, in which:

$$h_{\text{ow}} = c_{\text{ow}} \max \{ ka - s_{\text{min}}, 0 \}, \tag{18}$$

that is, $h_{\text{ow}}$ depends linearly on the wave steepness $ka$ and overwash occurs only if $ka$ exceeds a limiting value $s_{\text{min}}$. This choice is motivated by laboratory observations analyzed in Part B, in which the wave steepness at the ice edge seems to provide a good measure of the occurrence and intensity of overwash, as well as by the results of Skene et al. (2015, 2018), who obtained an approximately linear dependence of $h_{\text{ow}}$ on $ka$ for small wave steepness, relevant for this study (in the laboratory setup in Part B the maximum $ka_0$ at the ice edge equaled $\sim 0.05$). As Skene et al. (2015) observed a superlinear dependence of $h_{\text{ow}}$ on $ka$ for larger $ka$, $h_{\text{ow}} \sim (ka - s_{\text{min}})^\gamma$ with $\gamma > 1$ might be more suitable over a wider range of conditions; however, we do not consider $\gamma \neq 1$ in this work. We are also fully aware that the overwash thickness depends on a number of other factors, including ice thickness and density (and thus freeboard), floe sizes and their related flexural motion, and wave characteristics. However, as already mentioned, lack of validation data makes more sophisticated parametrizations unsupported. In computations in Part B, equation (17) is used with $h_{\text{ow}}$ computed from (18) and $s_{\text{min}}$, $c_{\text{ow}}$ treated as adjustable parameters that might be different for different floe sizes. It is assumed that the energy flux $\dot{E}_{\text{ow},i,i+1}$, occurring locally at the boundaries between ice floes, describes the energy of the propagating wave "removed" between floe $i$ and $i+1$.

### 2.4 Numerical algorithm

In the model described in sections 2.1–2.3, the wave energy dissipation $S_{\text{sd}}$ is computed based on the relative ice–water velocity $u_{\text{rel}}$ integrated over several wave periods. In turn, computation of $u_{\text{rel}}$ requires running the DEM with spatially variable (and known) wave amplitude as input. Analogous interdependencies occur in the computation of wave attenuation due to overwash. As we are interested in a quasi-stationary state, in which the floes move and collide with their neighbors, but the wave amplitude does not change in time, we use an iterative algorithm. The model is initialized with wave amplitude $a_i$ at each floe equal to the specified incident amplitude $a_0$. Then the following steps are repeated until the solution converges:

1. The model is run over $n_0$ wave periods to reach a stationary state.




2. Over the next $n_T$ wave periods, $S_{\mathrm{sd}}$ is computed for each floe using (13)–(15). Numerically, for rectangular floes considered here:

$$S_{\mathrm{sd}} = -\frac{\rho_w C_{sd}}{n_T n_t n_x} \sum_{j=1}^{n_T n_t} \sum_{k=1}^{n_x} u_{\mathrm{rel},j,k}^3, \tag{19}$$

where $L_x = n_x \Delta x$ and $T = n_t \Delta t$, $n_t$ and $n_x$ are integers, $\Delta t$ is the time step of the model, and $\Delta x$ the spatial resolution in the wave propagation direction.

3. New wave amplitude $a_i$ at the center of the $i$-th floe ($i = 2, \ldots, N_f$) is computed from (12) and from the amplitude $a_{i-1}$ at the center of floe $i-1$, assuming that the dissipation equals $S_{\mathrm{sd},i-1}$ over a distance between $x_{i-1}$ and $x_{i-1} + r$, and $S_{\mathrm{sd},i}$ over a distance $x_i - r$ and $x_i$ (if there is open water between floes $i-1$ and $i$, dissipation there is zero):

$$a_i = \max\left\{ \left[ a_{i-1}^2 + \frac{L_x}{\rho_w g c_g} (S_{\mathrm{sd},i-1} + S_{\mathrm{sd},i}) \right]^{1/2}, 0 \right\}. \tag{20}$$

4. If overwash effects are taken into account, $h_{\mathrm{ow},i}$ is computed from (18) for each floe, and $a_i$s are updated based on (17):

$$a_i = \max\left\{ \left[ a_{i-1}^2 - 3g^{1/2} \frac{h_{\mathrm{ow}}^{5/2}}{c_g} \right]^{1/2}, 0 \right\}. \tag{21}$$

The convergence criterion is based on the maximum wave amplitude difference between two consecutive loops of the algorithm: $\max_i\{|a_{i,\mathrm{old}} - a_{i,\mathrm{new}}|/a_{i,\mathrm{old}}\} < \delta$, where $\delta$ is set by the user.

In the present model version, when computing $u_{\mathrm{rel}}$ in (14), the same amplitude is used over the entire floe length – which is equivalent to an assumption that wave energy attenuation per ice floe is not very large. This assumption makes the attenuation algorithm consistent with the rest of the model (e.g., the $F_{\mathrm{w}}$ force is computed for constant $a$ for each floe).

It is worth noting that the number of iterations necessary for convergence increases with the distance over which attenuation is computed – as each location is affected by the situation in the up-wave direction, the convergence criterion is reached very fast close to the ice edge and the required number of iterations increases with increasing $x$. Not surprisingly, the model converges more slowly with higher restitution coefficients $\varepsilon$ and higher drag coefficients $C_{sd}$, i.e., more energetic collisions and stronger ice–water coupling.

## 3 Special case of ice concentration $c = 1$

Before proceeding to an analysis of full DEM simulations with collisions, it is useful to consider a limiting case with ice concentration $c = 1$ and horizontally confined ice, i.e., when $u_i(x,t) = 0$. In this case, $u_{\mathrm{rel}} = |u_{w,0}|$ and, from (3), its phase-averaged third power:

$$\overline{u_{\mathrm{rel}}^3} = \frac{4}{3\pi} \left( \frac{a\omega}{\tanh[kh]} \right)^3, \tag{22}$$





so that the wave attenuation can be computed analytically from the set of equations formulated in section 2.3. We have (disregarding overwash effects):

$$a\frac{da}{dx} = \frac{S_{\mathrm{sd}}}{\rho_w g c_g} = -\frac{C_{\mathrm{sd}}}{g c_g}\overline{u_{\mathrm{rel}}^3}, \tag{23}$$

which leads to:

$$\frac{da}{dx} = -\alpha_c a^2 \quad \text{with} \quad \alpha_c = \frac{4C_{\mathrm{sd}}}{3\pi g}\frac{\omega^3}{c_g \tanh^3[kh]}. \tag{24}$$

The index 'c' in the attenuation coefficient $\alpha_c$ should indicate that it represents a limiting case of confined ice, with no ice motion and thus no collision effects.

Notably, equation (24) has the form (1) discussed by Squire (2018), with $n = 2$. The solution of (24) is:

$$\frac{a(x)}{a_0} = \frac{1}{a_0 \alpha_c x + 1}. \tag{25}$$

The attenuation is non-exponential and, not surprisingly, $\alpha_c$ increases linearly with $C_{sd}$. Importantly, $\alpha_c$ is also frequency-dependent through the term $\omega^3/(c_g\tanh^3[kh])$. Thus, it is also directly dependent on the dispersion relation used. In the general case of the full elastic plate model (6), (7):

$$\frac{\omega^3}{c_g \tanh^3[kh]} = \frac{2\omega^4}{g}\frac{A^2}{\left[B\left(1 + \frac{2kh}{\sinh[2kh]}\right) + 4A(B-1)\right]\tanh^4[kh]}, \tag{26}$$

where, for the sake of brevity, we introduced the notation $A = 1 + \beta_1 k \tanh[kh]$ and $B = 1 + \beta_2 k^4/g$. In the simplest version of this model, i.e., when open-water dispersion relation is assumed, $A = B = 1$ and:

$$\alpha_{c,\mathrm{ow}} = \frac{8C_{\mathrm{sd}}}{3\pi g^2}\tilde{f}(kh)\omega^4, \quad \text{with} \quad \tilde{f}(kh) = (\tanh[kh])^{-4}\left(1 + \frac{2kh}{\sinh[2kh]}\right)^{-1}. \tag{27}$$

Thus, in deep water, when $\tilde{f}(kh) \to 1$, $\alpha_{c,\mathrm{ow}}$ is proportional to $\omega^4$ (note that the attenuation coefficient in this case differs from that obtained by Kohout et al., 2011, only by a constant, as they used the peak orbital velocity instead of phase-averaged velocity to compute $S_{\mathrm{sd}}$). In more general conditions of finite water depth, $\alpha_{c,\mathrm{ow}}$ has an $\omega^4$-tail (see black curves in Fig. 1b).

In the case of the mass loading model, $A > 1$ and $B = 1$, so that:

$$\alpha_{c,\mathrm{ml}} = A^2 \alpha_{c,\mathrm{ow}}, \tag{28}$$

and, as $A$ itself is an increasing function of $\omega$ (through its dependence on $k$), the mass loading model predicts faster-than-$\omega^4$ increase of $\alpha_{c,\mathrm{ml}}$ with $\omega$ (red and violet curves in Fig. 1b; note also that, for given $h_i$, the mass loading model produces positive group velocities only for $\omega^2 < g\rho_w/(\rho_i h_i)$). The difference between $\alpha_{c,\mathrm{ml}}$ and $\alpha_{c,\mathrm{ow}}$ becomes larger with increasing ice thickness $h_i$. In deep water, $\alpha_{c,\mathrm{ml}} \sim (1 + \rho_i/\rho_w k h_i)^2$, but due to typically very small values of $k h_i$, the relationship between $\alpha_{c,\mathrm{ml}}$ and $h_i$ can be regarded as approximately linear (as observed, e.g., by Doble et al., 2015).

If $\beta_2 > 0$, i.e., $B > 1$, the rate of increase of $\alpha_c$ with $\omega$ slows down relative to the open water model (blue and yellow curves in Fig. 1b). In this general case, expression (26) cannot be written in the form $\tilde{c}\omega^m$ in the whole frequency range. However, the



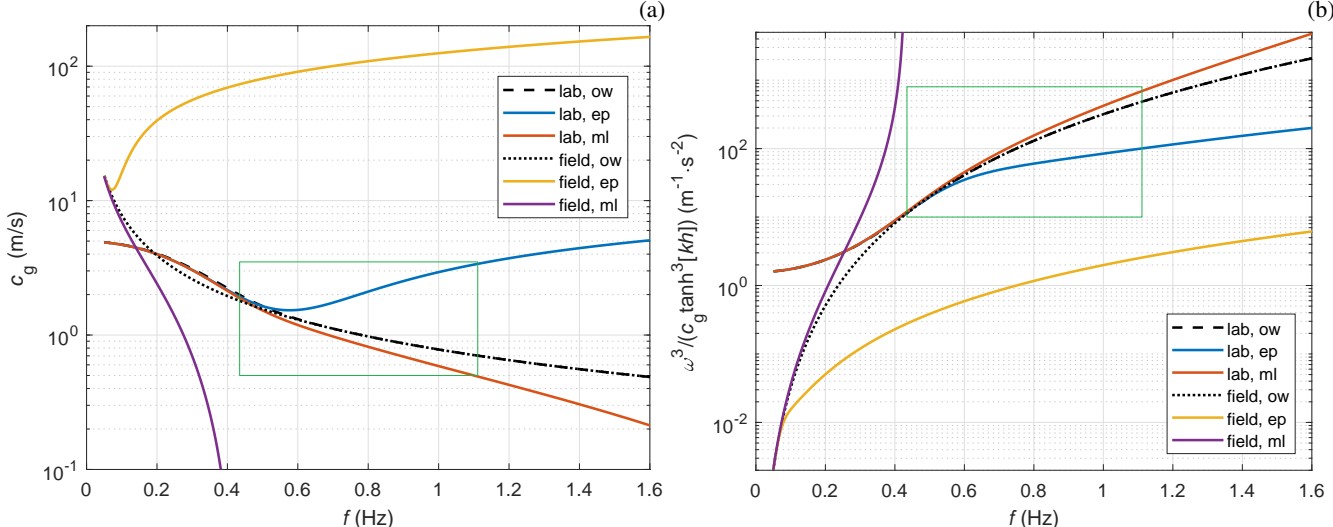

**Figure 1.** Group velocity $c_g$ (a) and the ratio $\omega^3/(c_g \tanh^3[kh])$ (b) versus wave frequency $f = \omega/(2\pi)$, computed for thin laboratory ice (test series 3000 from the LS-WICE experiment analyzed in Part B) and for field conditions with $h = 1000$ m, $\rho_w = 1025$ kg/m³, $\rho_i = 910$ kg/m³, $E = 6 \cdot 10^9$ Pa, $h_i = 1.5$ m. The corresponding open water solutions are shown in black (note that the two black curves overlap for $f > 0.5$ Hz). Green rectangles mark regions covered by the LS-WICE experiment.

high-frequency tail of $\alpha_{c,\mathrm{ep}}$ can be well approximated in this form. For the two examples shown in Fig. 1, the least-square fit of a $\tilde{c}\omega^m$ function to the data gives $m = 1.994$ and $m = 2.397$ for the laboratory and field-scale case, respectively.

This very different behavior of $\alpha_c(\omega)$ in the mass-loading and elastic-plate models (originating from the group velocity decreasing or increasing with wave frequency, respectively; Fig. 1a) indicates that one should expect very different wave attenuation patterns related to ice–water drag in ice composed of small and large floes. Differences in dispersion relation will lead to differences in attenuation rates, with very strong damping of high-frequency waves in fields of small ice floes (for which the mass-loading model is a good approximation), and with roughly $\omega^2$–$\omega^{2.5}$ damping in continuous ice or fields of large ice floes. We return to this fact in the discussion section.

## 4    Modelling results

### 4.1    Model setup

We set up the DEM model for conditions corresponding to those from LS-WICE series 3000 (see Part B). The ice sheet is 42 m long, and three floe lengths $L_x$ are considered: 0.5 m (number of floes $N_f = 84$), 1.5 m ($N_f = 28$), and 3.0 m ($N_f = 14$). For each floe size, the model is run for several different combinations of the following parameters: wave period $T$ (1.1, 1.2, 1.4, 1.5, 1.6, 1.8, 2.0 s), incident wave amplitude $a_0$ (0.0125, 0.015, 0.02, 0.025 m), drag coefficient $C_{sd}$ (0.005, 0.01, 0.05, 0.1, 0.15, 0.2), restitution coefficient $\varepsilon$ (0.2, 0.4, 0.6, 0.8) and initial floe–floe distance $d_f$ (0.005, 0.010, 0.020, 0.050 m). In each





model run, the floes are initially placed along the $x$ axis such that $x_1 = L_x/2$ and $x_{i+1} = x_i + L_x + d_f$ for $i = 2, \ldots, N_f$ (tests with random initial locations of the floes have shown that this aspect of the setup has no influence on the results). Additionally, for each value of $T$ three values of wavenumber $k$ and group velocity $c_g$ were considered, computed from the EP, OW and ML dispersion relations. Thus, the parameter space considered has 7 dimensions.

As described in Part B, the ice in LS-WICE was constrained horizontally by a floating boom and a sloping beach. In DEM, an analogous effect is obtained by adding a linear spring force $F_s$ to the first and last floe, with $F_{s,i}(t) = k_s(x_i(t) - x_i(0))$ for $i = 1$ and $i = N_f$. The value of the spring constant $k_s$ was set to $9 \cdot 10^4$ N·m$^{-1}$ (tests showed that the value of $k_s$ doesn't have visible results on the simulated attenuation, of interest in this study).

The time step $\Delta t$ used in the simulations equaled $1 \cdot 10^{-4}$ s. In the algorithm (see Section 2.4) $\delta = 10^{-3}$ was used in the
convergence criterion, with $n_0 = 10$ and $n_T = 5$. The spatial resolution in numerical integration of dissipation, used in equation (19), was $\Delta x = 0.01$ m.

As already mentioned, the analysis presented in the remaining parts of this paper concentrates on the role of ice–water drag, i.e., it is limited to results obtained without overwash, $S_{ow} = 0$. Simulations with overwash are discussed in Part B.

## 4.2   Influence of the model parameters on simulated wave attenuation

We begin exploring the model behavior with an analysis of the influence of the restitution coefficient $\varepsilon$ on wave attenuation. Obviously, by definition of $\varepsilon$, the lower its value the higher the fraction of kinetic energy of colliding objects that is dissipated during collisions. However, these energy losses, directly affecting the motion of the ice, do not automatically lead to the attenuation of the energy of the waves. To the contrary, as Fig. 2 clearly shows, the higher the $\varepsilon$, the lower the wave amplitude. The mechanism behind this relationship, described by Herman (2018) and mentioned in the introduction, is related to enhanced
relative ice–water velocities after collisions, leading to enhanced stress and thus stronger dissipation of wave energy.

Another aspect of the results immediately seen in Fig. 2 is that in most cases the slope of the $a(x)$ curve changes with distance from the ice edge: $da/dx$ is large close to the ice edge, within a relatively narrow zone of very strong attenuation, and becomes smaller further downwave. This effect is related to the rearrangement of the mean positions of the floes within the space available to them. As in every forced granular gas, the "atoms" tend to disperse from regions with higher granular
pressure to regions where the granular pressure is lower. Thus, close to the ice edge, where collisions are more energetic due to stronger forcing (higher wave amplitudes), the local ice concentration becomes slightly lower and the floes accumulate further downwave, in a densely packed zone of ice concentration close to 100%, i.e., with floes in permanent contact with their neighbors (Fig. 3a). The width of the collisional zone at the ice edge decreases with increasing $\varepsilon$, and the above-mentioned change of slope of the $a(x)$ curve corresponds to the location of the boundary between those two regions (see colour dots in
Figs. 2 and 3a). The two zones are, not surprisingly, characterized by different balance of forces. In the compact region with permanent floe–floe contact, the wave-induced forces are balanced by the contact forces, with drag force roughly two orders of magnitude lower (Fig. 3b–d); close to the ice edge, phase-averaged ice–water drag is still lower than the remaining forces, but it contributes a significantly larger part to the overall force balance. All these differences between the two regions are clearly seen in the time series of the energy dissipation term $S_{sd}$ (Fig. 4). For floes close to the ice edge, large spikes in $S_{sd}$ occur regularly





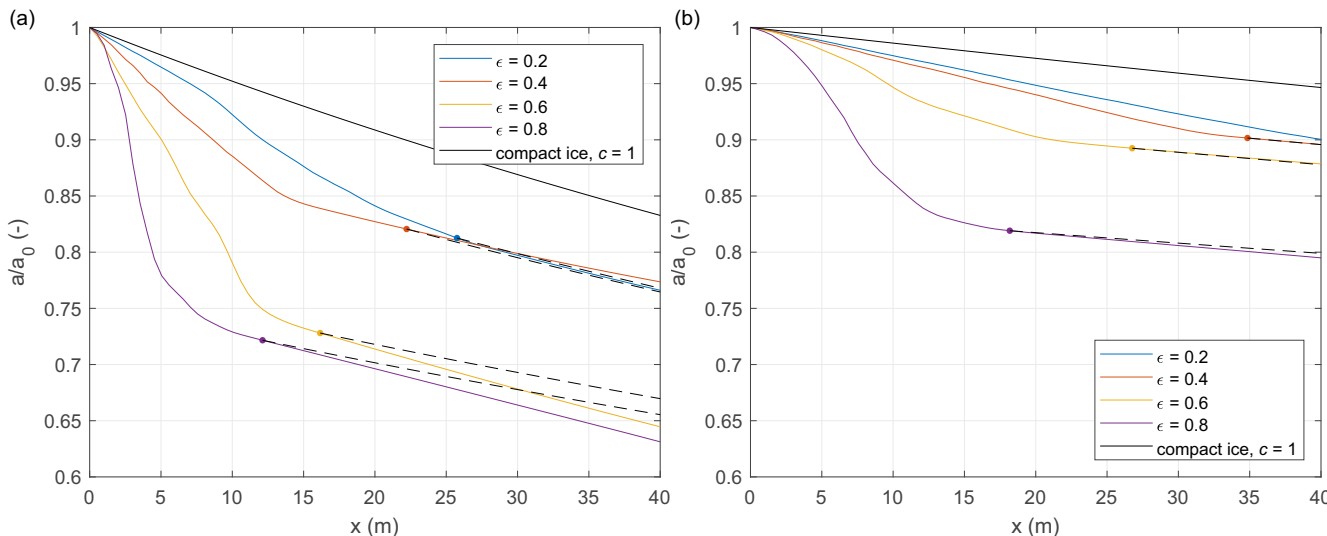

**Figure 2.** Computed relative wave amplitude $a/a_0$ for simulations with $L_x = 0.5$ m, $T = 1.2$ s, $a_0 = 0.0125$ m, $d_f = 0.005$ m, $C_{\mathrm{sd}} = 0.05$, with 'ML' (a) and 'EP' (b) dispersion relation. Colors correspond to different restitution coefficients $\varepsilon$, continuous black line shows the curve computed from equation (25) with $\alpha_c$ from (24). Dots mark locations where the phase-averaged floe–floe distance drops below $10^{-4}$ m (see Fig. 3a), and dashed black lines originating from those points show corresponding solutions for compact ice.

after each collision. Floes far from the ice edge experience very low, periodically varying $S_{\mathrm{sd}}$ related to small displacements from their average positions. Between those two regions of relatively regular – collisional or non-collisional – motion, the floes experience irregular fluctuations of their mean position (not shown) and associated periods with higher and lower collision rates, in effect producing erratic temporal patterns of $S_{\mathrm{sd}}$ (red curve in Fig. 4). Coming back to the wave attenuation, it is not

5    surprising that the simulated attenuation rates in the downwave high-concentration region are very close to those computed analytically for motionless ice (dashed lines in Fig. 2).

It is also worth noting that the existence of the collisional zone at the ice edge, producing strong attenuation, is directly related to the fact that the ice edge position is fixed in space – by the boom in the laboratory, and by the additional spring force in the model. Without that force, the floes drift gradually in the upwave direction (again, towards lower granular pressure) until

10    the ice concentration drops sufficiently so that collisions become sporadic. We return to this issue in the discussion section.

As can be expected from the analysis in Section 2.3, the dispersion relation has a very strong influence on the simulated attenuation rates (compare panels a,b in Fig. 2). With all other model parameters equal, 'EP' dispersion relation will always lead to lower attenuation rates than 'ML'. Thus, at least two mechanisms contribute to stronger attenuation when ice floes are small. First, dispersion in ice fields composed of small floes is better described by the 'ML' than by the 'EP' model. And

15    second, small floes undergo more vigorous collisions, with larger instantaneous accelerations and more collisions per distance travelled by the wave. In an example shown in Fig. 5, the 'ML' model is likely more suitable for small floes with $L_x = 0.5$ m,

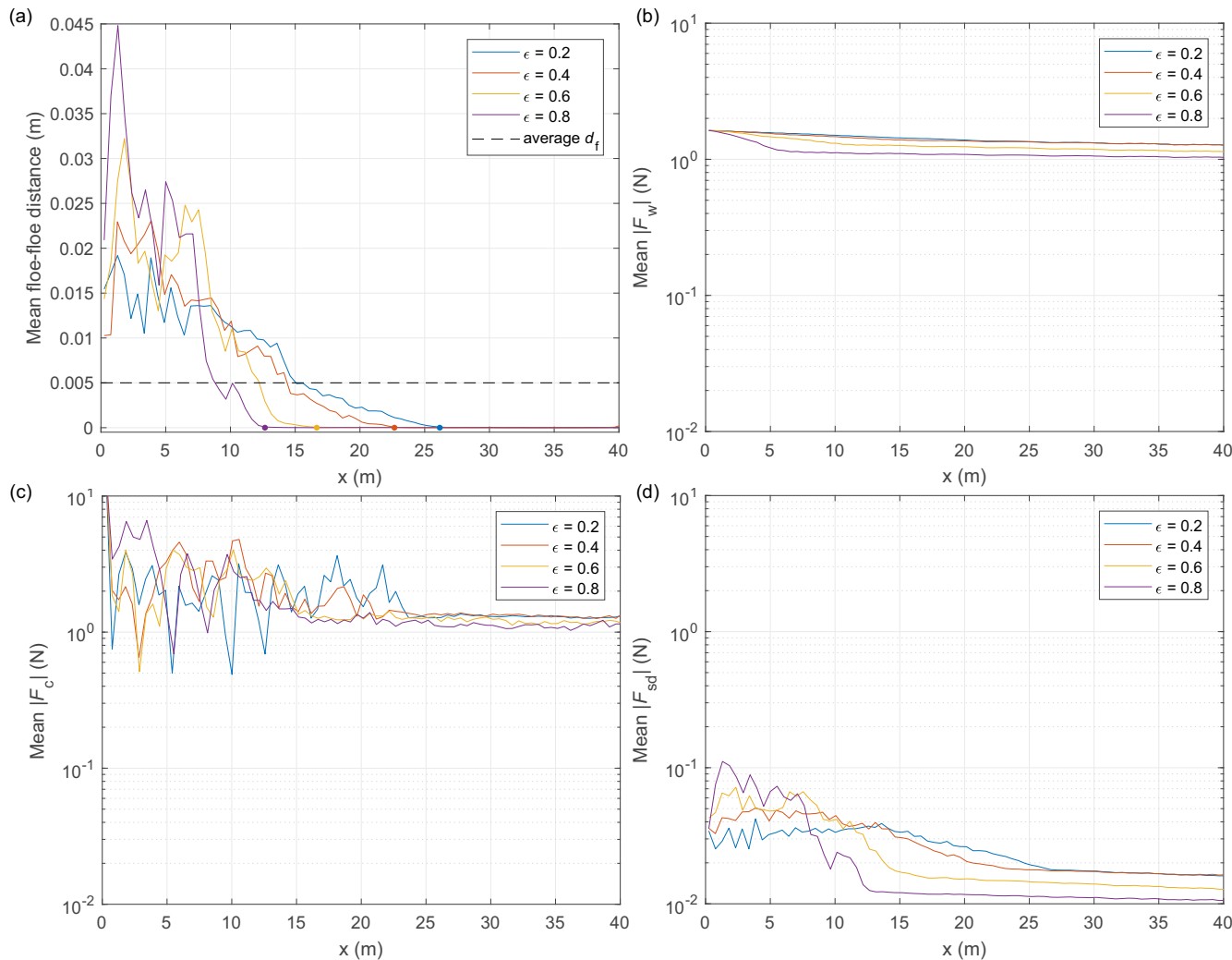

**Figure 3.** Results of simulations with $L_x = 0.5$ m, $T = 1.2$ s, $a_0 = 0.0125$ m, $d_f = 0.005$ m with ML dispersion relation (as in Fig. 2a): mean floe–floe distance (a), mean $|F_w|$ (b), mean $|F_c|$ (c), and mean $|F_{sd}|$ (d) for four values of restitution coefficient $\varepsilon$. In (a), black dashed line shows the domain average $d_f$, and dots mark locations, where the average floe–floe distance drops below $10^{-4}$ m.

and the 'EP' model more suitable for large floes with $L_x = 3.0$ m, so that the expected difference in attenuation observed in these two situations can be as large as between the dashed yellow and the continuous blue line in Fig. 5.

The fact that the frequency and character of collisions play a crucial role in shaping floe dynamics and wave energy dissipation in the region close to the ice edge means that the ice concentration, and thus the floe–floe distances, should have a visible influence on attenuation. This is indeed the case (Fig. 6a): when $d_f$ decreases, attenuation increases. However, as can be seen for the results with short waves, stronger attenuation close to the ice edge means that the zone of strong attenuation becomes narrower, so that further downwave the relationship between $d_f$ and $a/a_0$ reverses (in Fig. 6a, no analogous effect is

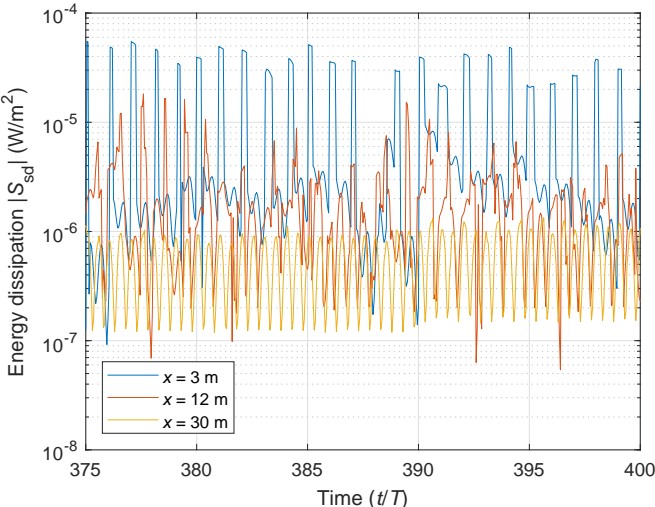

**Figure 4.** Time series of the modulus of the wave energy dissipation term $|S_{\mathrm{sd}}|$ in simulations with $L_x = 0.5$ m, $T = 1.2$ s, $a_0 = 0.0125$ m, $C_{sd} = 0.05$, $\varepsilon = 0.6$ and 'ML' dispersion relation (see yellow curve in Fig. 2a), for three selected floes, located at 3 m, 12 m and 30 m from the ice edge.

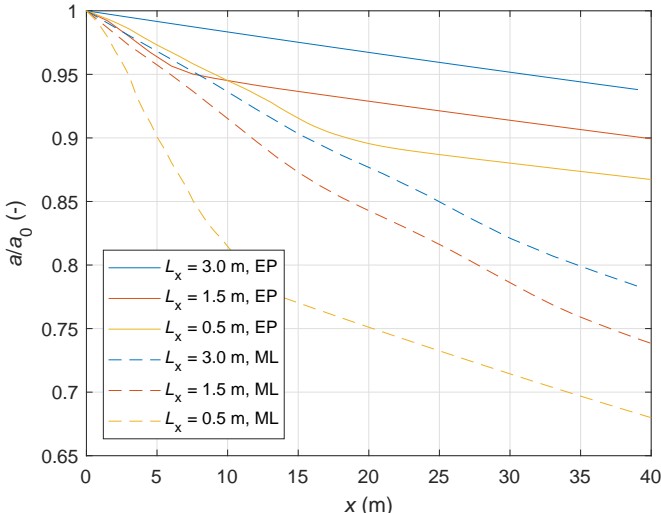

**Figure 5.** Computed relative wave amplitude $a/a_0$ for simulations with different floe sizes $L_x$ (colours), with $T = 1.4$ s, $a_0 = 0.015$ m, $d_f = 0.005$ m, $C_{\mathrm{sd}} = 0.05$, $\varepsilon = 0.6$, with 'EP' (continuous lines) and 'ML' (dashed lines) dispersion relation.

present for the longer waves with $T = 1.8$ s, because the collisional zone extends in this case over the whole model domain). Those examples illustrate how difficult it might be to "reconstruct" the attenuation curves from measurements available only at a limited number of locations (as in the case discussed in Part B), and how careful one should be when interpreting that kind of data.





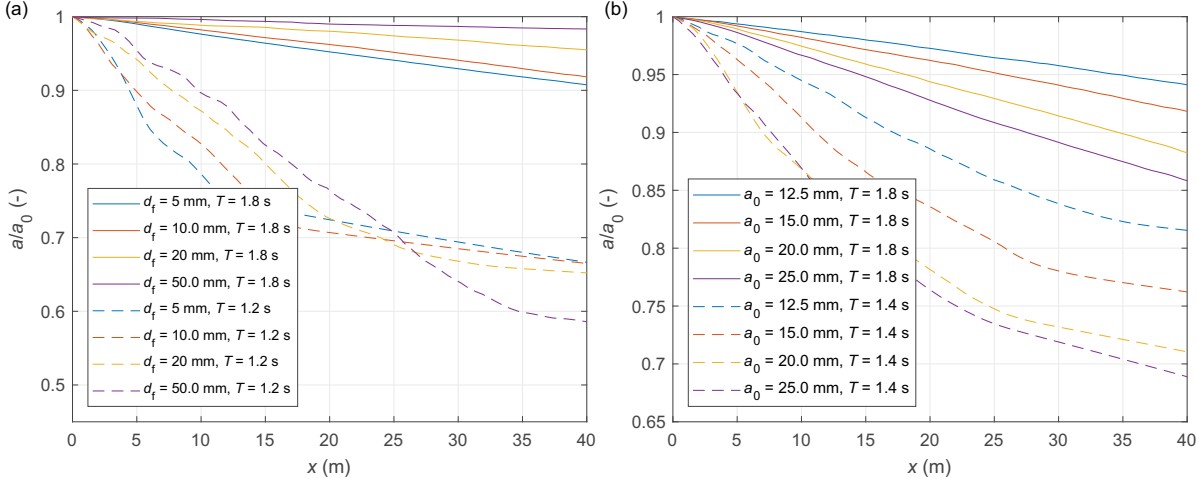

**Figure 6.** Computed relative wave amplitude $a/a_0$ for simulations with different average floe–floe distance $d_f$ (a) and different incident wave amplitude $a_0$ (b), for $L_x = 0.5$ m, $a_0 = 0.015$ m, $d_f = 0.005$ m, $C_{\mathrm{sd}} = 0.05$, $\varepsilon = 0.6$, for two different wave periods.

Finally, it is worth stressing that the modelled wave attenuation in both regions is strongly dependent on the incident wave amplitude $a_0$ (Fig. 6b). In the collisional zone at the ice edge, the wave amplitude decides on the surge amplitude of the floes and thus on the occurrence and intensity of collisions. Further downwave, the $a(x)$ curve is well described by equation (25), i.e., the attenuation rate is close to $a_0\alpha_c$.

## 5   Discussion and conclusions

As noted recently by Meylan et al. (2018), the dependence of the attenuation rate $\alpha$ on wave frequency follows directly from the formulation of a given model and therefore – if the model does not properly reproduce the relevant processes – its coefficients can be tuned to the observed attenuation at one frequency only. Reversing this argument, the observed functional forms of $a(x)$ and $\alpha(f)$ can be treated as signatures of physical attenuation processes that have shaped them. It is thus crucial to improve our understanding of how different attenuation mechanisms influence $a(x)$ and $\alpha(f)$. In this work, we concentrated on one of those mechanisms: dissipation of wave energy due to ice–water drag. We used DEM simulations and, in a limiting case of compact sea ice, an analytical analysis, in order to investigate how ice–water drag influences the dynamics of sea ice floes and the corresponding attenuation of wave energy. Several aspects of the results, mentioned in the text, are worth further discussions.

The DEM simulations predict a very distinctive pattern of wave attenuation resulting from combined effects of ice–water drag and collisions between ice floes. The results suggest that intense collisions between ice floes can be expected to occur only within a narrow zone close to the ice edge, which is also a zone of lowered ice concentration and of very strong attenuation – provided that the floes are not able to drift in the upwave direction. In natural conditions, forces keeping the ice edge in place may include compressive stress caused by wave reflection from the ice edge, as well as wind and/or average currents





with sufficient velocity so that the forces exerted by them on the ice compensate those related to increased granular pressure. It is interesting to notice that the elevated granular pressure can be sustained only by a constant energy input from the waves; otherwise, inelastic floe–floe collisions would lead not to increased, but to decreased collision rates. This makes the situation very different from the wind-forced sea ice studied by Herman (2011, 2012), where floes tended to accumulate in regions of

intense collisions, producing clusters with high ice concentration. In the present case, thanks to the interplay with wave forcing, the same basic mechanisms lead to the formation of the two zones described in section 4.2, with very different wave attenuation rates and collision patterns. Notably also, if the ice floes are small relative to the wavelength, very different attenuation should be expected in situations with confined ice edge (strong ice–water drag due to floe collisions at high ice concentration) and in situations with "free" ice edge (no collisions due to lowered ice concentration, floes able to follow the motion of the water).

The fact that the floes tend to accumulate in the inner zone, forming a semi-continuous ice cover with ice concentration close to 100% and limited horizontal ice motion, means that – if dissipation due to ice–water drag is significant – the expected attenuation rates within that zone should be close to those computed analytically in section 3. From a practical point of view, it substantially simplifies the situation, eliminating from the set of relevant variables those related to collisions. Crucially, as illustrated in section 3, the behaviour of $\alpha(\omega)$ in this case depends very strongly on the dispersion relation, with much weaker

dissipation in sea ice composed of large ice floes, behaving as elastic plates, and stronger dissipation in sea ice composed of small floes, behaving as rigid "mass points". It must be stressed here that the strong influence of the dispersion relation on $\alpha(\omega)$ is not limited to the dissipation mechanisms discussed in this paper. As the left-hand-side of equation (9) has the form $\mathrm{d}(c_g E_w)/\mathrm{d}x$, the value of $c_g^{-1}$ will always influence the energy attenuation, contributing to stronger attenuation in small floes (when $c_g$ is relatively low and decreases with increasing wave frequency) than in large floes (when $c_g$ is larger and

increases with increasing frequency; Fig. 1a). In many studies these effects are not taken into account and open-water dispersion relation is assumed (e.g., Meylan et al., 2018), although several observations, including those analyzed by Liu and Mollo-Christensen (1988) or Sutherland and Rabault (2016), show the influence of floe size on wave propagation speed (in the LS-WICE experiment discussed in Part B, for which the present DEM was configured, decreasing wavenumbers with increasing floe size were observed, as analyzed by Cheng et al., 2018). The example of attenuation due to ice–water drag, discussed in

this work, suggests that even small changes of $c_g$ may lead to noticeable changes of $\alpha$. In the inner zone far from the ice edge, where the floes tend to be larger and therefore $\alpha$ should be close to $\alpha_{c,\mathrm{ep}}$, the present model predicts a power-law tail in the relation $\alpha(\omega)$ with the power $m$ typically between 2 and 2.5 depending on ice properties, i.e., substantially lower than $m = 4$ for the open water dispersion relation. This is in agreement with many observations, although, obviously, it does not mean that the analyzed mechanism is significantly contributing to attenuation in real sea ice.

It is also worth noting that the influence of ice–water drag on wave energy attenuation depends very strongly on the drag law used. If, for example, a linear drag law $\tau_w \sim u_{\mathrm{rel}}$ is used instead of the quadratic law (15), exponential attenuation $a(x) = a_0 \exp[-\alpha_{c,l}x]$ is obtained instead of (25), with $\alpha_{c,l}$ proportional to $\omega^2/(c_g \tanh^2[kh])$, i.e., the increase of the attenuation coefficient $\alpha_{c,l}$ with $\omega$ is slower than predicted by the model described earlier. This illustrates that both the shape of the attenuation curve $a(x)$ and the attenuation coefficients are very sensitive to the formulation of the dissipation term in the

energy transport equation (9). On the other hand, any model with a dissipative force quadratically dependent on relative ice–





water velocity will exhibit a similar behaviour. For example, replacing the drag coefficient $C_{\mathrm{sd}}$, here representing skin drag, with a form drag coefficient, and replacing integration over the bottom surface of the floes in (13) with integration over their vertical walls, should not change the general attenuation behaviour described above.

A very important limitation of the model used here is the fact that it takes into account only the transmitted propagating
component ($T_0$ in the notation of Kohout et al., 2007) of the wave motion. As our analysis in Part B shows, the contribution of $T_0$ to the total wave amplitude in the LS-WICE experiment is variable and strongly dependent on the floe size/wavelength ratio. From the point of view of the ice–water drag, discussed in this paper, it is important to keep in mind that the additional modes – especially the propagating damped modes ($T_{-2}$, $T_{-1}$, $R_{-2}$, $R_{-1}$), which might have amplitudes comparable with $T_0$ – modify the spatial and temporal variability of $u_w$, thus modifying the instantaneous and phase-averaged $u_{\mathrm{rel}}$ and $S_{\mathrm{sd}}$. It
remains to be investigated how large those changes might be in different conditions. Moreover, the damped modes increase the water velocities close to the edges of the floes, as well as the amplitude of the vertical motion of floes' edges (the total amplitude is a sum of the amplitude of the propagating components, constant over the length of the floe, and the amplitude of the damped components, decreasing from floe edges towards its inner parts). Thus, the presence of the damped modes might modify overwash and, combined with floe collisions, contribute to the enhancement of turbulent mixing at floe boundaries.
Although analyzing interrelationships between those processes in full detail will require much more advanced models and observations, an initial step in that direction can be done by extending the present DEM model so that more realistic wave forcing can be used.

*Code availability.* The code of the DESIgn model is freely available at https://herman.ocean.ug.edu.pl/LIGGGHTSseaice.html and as a supplementary material to Herman (2016). The extended code necessary to reproduce the results presented in this paper, together with input
scripts, can be obtained from the corresponding author.

*Author contributions.* All authors contributed to planning of the research and to discussion of the results. A.H. developed the numerical model, performed the simulations and wrote the text.

*Competing interests.* The authors declare no competing interests

*Acknowledgements.* The development of the numerical model used in this work has been financed by the Polish National Science Centre
research grant No. 2015/19/B/ST10/01568 ("Discrete-element sea ice modeling – development of theoretical and numerical methods"). Coauthors SC and HHS are supported in part by ONR grant No. N00014-17-1-2862.



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
