# Peer review of "Wave energy attenuation in fields of colliding ice floes. Part A: Discrete-element modelling of dissipation due to ice-water drag"

_The Cryosphere, 2019_

## Referee Comment (RC1) · Anonymous Referee #1 · 18 Jul 2019

**1   General comments**

This paper is concerned with how ocean waves attenuate as they propagate in fields of sea ice, specifically how dissipation due to ice-water drag contributes to the attenuation. It represents Part A of two papers, the first focussed primarily on modelling and the second (Part B) focussed on experiments in a wave flume. I have been asked to review both papers. Respecting the journal's page limits—which don't appear to be immutably adhered to based on other published papers—a single longer paper combining Parts A and B would have made my assignment easier and have been preferable, because it is difficult to review Parts A and B interdependently as the manuscripts in-

tersect significantly. (I also conjecture that a synthesis of Parts A and B would likely have occupied less pages than Part A and Part B published independently because of repetition.) Be that as it may, I shall do my best.

Fortunately, I support publication of Part A (and Part B) subject to *very* minor amendments and remark that most of my specific comments below are intended to clarify and hopefully improve the manuscript rather than being obligatory.

Although the work considers a subset of the many potential dissipative mechanisms that are possible when waves traverse sea ice, which collectively depend on the multifaceted relationship between the incoming waves and the ice morphology, the paper should be read in the context of our poor overall understanding of the topic. While good progress has been made in accounting for the conservative redistribution of wave energy that arises because of scattering by and between ice floes, we know appreciably less about the physics of how energy is systematically removed from the waves as they travel through sea ice. Few models exist and historical data sets are perfunctory due to the sensitivity of the dissipative processes to the physical and mechanical properties of the ice cover and the attributes of the waves themselves. In sum, as the authors point out, "interpretation of the observed attenuation rates is extremely difficult, as it would require simultaneous measurement of several wave and ice characteristics over large distances."

The authors use a one-dimensional discrete-element model (DEM) to simulate the wave-induced surge and collisions of ice floes, coupled to the wave energy transport equation via phase-averaged source terms. The modest size of the ice floes and the wavelengths are similar, as details about the sea ice (and waves) were chosen to emulate a series of wave flume experiments reported in Part B that took place at the Hamburg Ship Model Basin. For a compact, horizontally-confined ice cover, the authors report a seemingly surprising result but one that is not without precedent, namely that nonexponential attenuation of wave amplitude occurs. They find that wave amplitude $a$ as a function of distance travelled $x$ behaves as $a(x) = 1/(\alpha x + 1/a_0)$, with $a_0 = a(0)$; an

example of the more general (nonlinear) power law attenuation hypothesized by Shen and Squire (1998) for pancake ice collisions triggered by proportionately longer waves (see also Squire, 2018) and, in fact, by Wadhams (1973) using an alternative creep parametrization. (In passing, I note that existing field observations do not allow power law attenuation to be straightforwardly tested against the pervasive $a(x) = a_0 \exp(-\alpha x)$ relationship for particular circumstances, because the usual confidence intervals associated with in situ wave measurements in ice covers are too large due to unavoidable experimental design constraints. Indeed, the authors assert, "In many cases, large scatter in observational data and/or limited number of measurement locations make the usage of more complicated models unjustified.") The attenuation rate, $\alpha$, depends on frequency $\omega$, i.e. how the waves disperse. The authors conclude that $\alpha \sim \omega^{2-2.5}$ for continuous ice, which does appear to be supported by observations. The DEM model predicts the existence of two zones, a narrow collisional zone with very strong attenuation near the ice edge and an interior zone where attenuation rates are less. While similar structures are frequently encountered in field data, e.g. see Squire and Moore (1980, doi: 10.1038/283365a0) and the explanation provided by the authors for the genesis of the outer zone seems plausible, other prospective causes exist.

**2   Specific comments**

1. While the theme of the paper is quite technical, the authors remind us that large scale sea ice models, e.g. Bateson et al. (2019)'s augmented CICE-based model coupled to a prognostic ocean mixed layer model, indicate that ice extent and volume are sensitive to ocean wave attenuation rates; waves break up the sea ice and move it around to a degree that is strongly dependent on how they are attenuated. Consequently, this paper is an important addition to the wave/ice interactions literature. To help the reader appreciate the significance of its outcomes, a little more could be said in regard to its relevance to climate change, as ocean

waves have unquestionably contributed to the demise of the Arctic summer sea ice by accelerating other effects such as ice-albedo feedback. It is unlikely that this is not also true in the Southern Ocean, given the intensity and uptrending of the wave climate there.

2. If, as the authors state, "[nonlinear] processes leading to the dissipation of wave energy take place within the ice itself as well as in the underlying water layer and include viscous deformation of the ice, overwash, vortex shedding and turbulence generation, friction between ice floes and between ice and water (form and skin drag), inelastic floe-floe collisions, breaking and rafting of floes, and many more," I personally believe it is unlikely that simple exponential attenuation via the linear differential equation $da/dx = -\alpha a$ will predominate. However, I also recognize that our ability to differentiate between it and a more sophisticated model such as $da/dx = -\alpha a^n$ may not be achievable because of the considerable uncertainty associated with most field data sets or remotely-collected data, e.g. satellite or airborne SAR, and that exponential decay may be perfectly reasonable for operational wave forecasting and large scale modelling projects. The authors could be a little more forthright in saying this, assuming that they agree with me.

3. In a similar vein, even if exponential attenuation is prevalent, it is very unlikely that a single attenuation coefficient $\alpha$ will hold, given the intense heterogeneity associated with most fields of sea ice floes. The authors could add words to this effect too.

4. While mostly saved for Part B, there is a statement on page 4 that even seemingly simple situations lead to wave propagation and attenuation that is shaped by several interrelated processes that are impossible to isolate from one another, and that several different model configurations can reproduce the observed attenuation rates. As the authors point out, this makes identification of processes actually responsible for dissipation a formidable task. This is a crucially important—

although a rather dispiriting and potentially controversial conclusion, which needs to come through much more strongly in the paper(s) than it currently does, as it potentially redirects the topic to a more empirical/statistical future. Again, I encourage the authors to be more assertive about this conclusion as, while it is mentioned early in the paper and is reiterated in Part B, it appears to be played down or absent in § 5 of Part A.

5. As noted above, the DEM model is one-dimensional. Not much is made of this but it may have serious implications that the authors need to discuss. While Part A includes only the transmitted mode, Part B assimilates more modes using the Kohout et al. (2007) analysis. In both cases all the energy is trapped in the wave vector direction. What is the effect of this?

6. One of the authors (Shen) has constructed an impressive, sophisticated model of sea ice to study wave propagation that includes a form of viscosity via a complex shear modulus. As such it is more general than the options provided by dispersion relation (6), which restricts us to open water, the mass-loading approximation and the elastic plate. There is also the simpler model of Robinson and Palmer (1990, doi: 10.1016/0022-460X(90)90661-I) that includes viscosity by means of a velocity-dependent damping term, and which seems to reproduce observations better in regard to the frequency dependence of $\alpha(\omega)$. Especially given that the mass-loading model tends not to be favoured by many of my colleagues, I ask the present authors to provide an explanation as to why they used dispersion relation (6). (Note that I am only asking for an explanation, not a reworking of the model.)

7. The authors point out the difference between the DEM in regard to the attenuation of wave energy and the Shen and Squire (1998) model, and they include source terms for both drag and overwash, the latter primarily targeted at the experiments reported in Part B and being especially important near the ice edge before waves have attenuated substantially. I understand the explanation for the difference but

what is the effect of choosing this particular formulation?

8. I really enjoyed the $c = 1$ analysis, especially the outcome that $da/dx = -\alpha_c a^2$, with its solution $a(x) = 1/(\alpha x + 1/a_0)$. This notwithstanding, I was bemused by why the limiting case of confined ice, i.e. with no collisions, should satisfy a particular exemplar of the more general differential equation predicted by Shen and Squire (1998) when collisions were modelled, viz. $da/dx = -\alpha a^n$ with $n = 2$. I guess this reinforces point 4 above, paraphrased that different physics can lead to the same outcome. I also enjoyed the subsequent analysis that investigated how $\alpha$ varied for the three different dispersion relations used, revealing $\omega^4$ behaviour for open water, $\omega^{>4}$ behaviour for the mass-loading approximation and $\omega^{<4}$ for the continuous elastic plate. The asymptotic dependence of $\alpha(\omega)$ is important because of the consistency of in situ field measurements that suggest a power between 2 and 3.

9. As hinted at earlier, I would contest the statement made at the end of §3 that the mass-loading model is a good approximation of waves propagating through small ice floes or, at least, ask the authors to provide a reference to that effect. However, I do agree with the sentiment expressed that, for this model, it is the dispersion relation (and $c_g$) that causes differences in how $\alpha$ behaves. Because, taken out of context, this is counter-intuitive as one expects dispersion relation (6) to provide information about how the principal propagating mode disperses rather than attenuates, the authors could explain this better.

10. The paper continues by using the model derived in earlier sections with some of the laboratory data from Part B, essentially embarking on a sensitivity study to see which parameters influence specific outcomes. Some results are not obvious, e.g. why an increased restitution coefficient should lead to lower wave amplitudes, but this is explained. A change of slope is very evident in many circumstances in Fig. 2, dividing the ice field into two zones of high and low attenuation as is often observed in the field (see Squire and Moore, 1980, e.g.). Comparable curves in Fig. 2a and 2b show the effect of the mass-loading and elastic plate dispersion relations. The figure is well explained and the point is well made that smaller floes will always lead to stronger attenuation because of at least two mechanisms. Floe size is investigated in Fig. 5, again with the elastic plate dispersion relation showing a smaller reduction in $a(x)$ than the mass-loading one, while average floe to floe distance is considered in Fig. 6a, where similar behaviour is evident save for very short waves where the zone of intense attenuation is particularly narrow. How the initial amplitude $a_0 = a(0)$ affects $a(x)$ is shown in Fig. 6b. In sum, the properties and the outputs of the model are well tested and well explained by a number of figures, of which I have mentioned just a few. The authors are congratulated for their thorough analysis.

11. The authors begin §5 on page 16 by reminding us that $a(x)$ and $\alpha(\omega)$ are signatures of the underlying dissipative physics; the question is can the behaviour of these parameters be used to shed light on that physics? We are told that "the DEM simulations predict a very distinctive pattern of wave attenuation resulting from ice-water drag and collisions," but, as the authors point out, conditions at a real ice edge may be dominated by compressive forces exerted by radiation stress, winds and currents compensating increased granular pressure within the ice cover sustained by the waves themselves. Furthermore, the wavelengths considered in the simulations are comparable to the size of the floes, which is different from the situation that is often seen in the winter Antarctic and more recently in the western Arctic where a pancake ice zone forms at the ice margin. What is particularly noteworthy—and deserving more comment to finish—is the result that the tail of $\alpha(\omega)$ follows a power law $\sim \omega^{2-2.5}$ within the ice interior, quite different from open water and close to what has been observed in situ.

**3 Technical corrections**

I have very few technical corrections that I ask the authors to address, as follows

1. Page 3, line 5. Squire (2018) is not really a review paper. Rather, that paper justifies and then fleshes out an explanation for the differential equation $da/dx = -\alpha a^n$ as a power-law fluid and tests the outcome of different values of $n$. The word *review* should be removed.

2. Page 3, line 13. The Part B paper is not in the bibliography.

---

## Referee Comment (RC2) · Anonymous Referee #2 · 5 Aug 2019

This paper deals with theoretical aspects relevant to the dissipation of waves traveling in sea ice. Particular focus is devoted to the role and parameterization of the ice-water drag as relevant dissipation mechanism. A discrete-element model (DEM) was employed in order to simulate the motion and collisions of the ice floes under the wave action, coupled to the wave energy transport with phase-averaged source terms. As the aim of the paper was to explain wave dispersion and attenuation observed in a wave channel, wave energy dissipation due to overwash was also considered for completeness. Indeed, laboratory wave data are the subject of a companion paper (Part B), for which I was also asked to review. Unlike the water-ice drag, a minor role was recognized to the overwash mechanism to account for wave energy dissipation. Wave

energy attenuation was analytically analyzed in the case of compact, horizontally confined ice cover. Interestingly, the authors show that a non-exponential wave attenuation law with the distance has to be expected if a quadratic drag law at the ice-water interface is assumed. Current wave field observations do not allow to discriminate between the widely accepted/assumed exponential wave energy decline against other types of wave attenuation as a result of the large data scatter provided by in situ wave measurements. This means that new technologies should be envisaged to overcome this experimental limit. Authors also show that the attenuation rate is frequency-dependent and the dependence is related to the dispersion relation used. To this end, the authors assumed a wave dispersion relationship which blends shortening (mass loading) and lengthening (elasticity) of the open sea wavelength proportionately to the nature and rheology of sea ice. I support this paper. Some specific comments will be reported below, which I would like to read in the final version of the paper: 1) a discussion to explain the choice of the wave dispersion (eq. 6) could be added. The reason is the presence of the mass loading term. The weak point is that it could not adequately represent the ice floes assumed in the paper in terms of horizontal size/ wavelength ratio. In fact, the mass loading term is considered valid for really point-like ice floes (compared to the wavelength). 2) The relevance of papers like this is the possibility to extrapolate to the real world what learned for the in-door environment, also in simulation. So, to what extent do the authors think their model formulation can represent the complexity of our changing Arctic and Antarctic MIZ?

---

## Author Comment (AC1) · 12 Sep 2019

**Response to the comments of Reviewer #1**

This paper is concerned with how ocean waves attenuate as they propagate in fields of sea ice, specifically how dissipation due to ice-water drag contributes to the attenuation. It represents Part A of two papers, the first focussed primarily on modelling and the second (Part B) focussed on experiments in a wave flume. I have been asked to review both papers. Respecting the journal's page limits—which don't appear to be immutably adhered to based on other published papers—a single longer paper combining Parts A and B would have made my assignment easier and have been preferable, because it is difficult to review Parts A and B interdependently as the manuscripts intersect significantly. (I also conjecture that a synthesis of Parts A and B would likely have occupied less pages than Part A and Part B published independently because of repetition.) Be that as it may, I shall do my best.

First of all, we are very grateful that the Reviewer agreed to review both parts of our paper, and we truly appreciate all comments and suggestions regarding the contents of both parts. We agree that, from the point of view of the reviewers, a single paper is much easier to handle. However, we are convinced that dividing our results into two parts is beneficial from the point of view of presentation of those results, in that the message from each stage of our study – the theoretical part, the numerical simulations, and the laboratory data analysis – can be clearly presented in sufficient detail. Our decision to split the manuscript was motivated not by the journal's page limit, but rather by the volume of the results. We believe that two separate parts make the message more clear for the reader.

Fortunately, I support publication of Part A (and Part B) subject to very minor amendments and remark that most of my specific comments below are intended to clarify and hopefully improve the manuscript rather than being obligatory.

**Thank you! We will do our best to incorporate your suggestions and comments into the new version of our manuscript.**

Although the work considers a subset of the many potential dissipative mechanisms that are possible when waves traverse sea ice, which collectively depend on the multifaceted relationship between the incoming waves and the ice morphology, the paper should be read in the context of our poor overall understanding of the topic. While good progress has been made in accounting for the conservative redistribution of wave energy that arises because of scattering by and between ice floes, we know appreciably less about the physics of how energy is systematically removed from the waves as they travel through sea ice. Few models exist and historical data sets are perfunctory due to the sensitivity of the dissipative processes to the physical and mechanical properties of the ice cover and the attributes of the waves themselves. In sum, as the authors point out, "interpretation of the observed attenuation rates is extremely difficult, as it would require simultaneous measurement of several wave and ice characteristics over large distances."

The authors use a one-dimensional discrete-element model (DEM) to simulate the wave-induced surge and collisions of ice floes, coupled to the wave energy transport equation via phase-averaged source terms. The modest size of the ice floes and the wavelengths are similar, as details about the sea ice (and waves) were chosen to emulate a series of wave flume experiments reported in Part B that took place at the Hamburg Ship Model Basin. For a compact, horizontally-confined ice cover, the authors report a seemingly surprising result but one that is not without precedent, namely that nonexponential attenuation of wave amplitude occurs. They find that wave amplitude a as a function of distance travelled x behaves as  $a(x) = 1/(\alpha x+1/a_0)$ , with  $a_0 = a(0)$ ; an example of the more general (nonlinear) power law attenuation hypothesized by Shen and Squire (1998) for pancake ice collisions triggered by proportionately longer waves (see also Squire, 2018) and, in fact, by Wadhams (1973) using an alternative creep parametrization. (In passing, I note that existing field observations do not allow power law attenuation to be straightforwardly tested against the pervasive  $a(x) = ao \exp(-\alpha x)$  relationship for particular circumstances, because the usual confidence intervals associated with in situ wave measurements in ice covers are too large due to unavoidable experimental design constraints. Indeed, the authors assert, "In many cases, large scatter in observational data and/or limited number of measurement locations make the usage of more complicated models unjustified.") The attenuation rate,  $\alpha$ , depends on frequency  $\omega$ , i.e. how the waves disperse. The authors conclude that  $\alpha \sim \omega^{2-2.5}$  for continuous ice, which does appear to be supported by observations. The DEM model predicts the existence of two zones, a narrow collisional zone with very strong attenuation near the ice edge and an interior zone where attenuation rates are less. While similar structures are frequently encountered in field data, e.g. see Squire and Moore (1980, doi: 10.1038/283365a0) and the explanation provided by the authors for the genesis of the outer zone seems plausible, other prospective causes exist.

**Specific comments**

1. While the theme of the paper is quite technical, the authors remind us that large scale sea ice models, e.g. Bateson et al. (2019)'s augmented CICE-based model coupled to a prognostic ocean mixed layer model, indicate that ice extent and volume are sensitive to ocean wave attenuation rates; waves break up the sea ice and move it around to a degree that is strongly dependent on how they are attenuated. Consequently, this paper is an important addition to the wave/ice interactions literature. To help the reader appreciate the significance of its outcomes, a little more could be said in regard to its relevance to climate change, as ocean waves have unquestionably contributed to the demise of the Arctic summer sea ice by accelerating other effects such as ice-albedo feedback. It is unlikely that this is not also true in the Southern Ocean, given the intensity and uptrending of the wave climate there.

We do hope, of course, that our paper "is an important addition to the wave/ice interactions literature". At the same time, as the Reviewer notices at the beginning of this comment, our work concentrates on details of selected processes accompanying wave propagation in sea ice, and the link to climate-related problems seems rather far-fetched. But we will consider adding some comments to the discussion section, describing why our results might be relevant from the polar climate perspective.

2. If, as the authors state, "[nonlinear] processes leading to the dissipation of wave energy take place within the ice itself as well as in the underlying water layer and include viscous deformation of the ice, overwash, vortex shedding and turbulence generation, friction between ice floes and between ice and water (form and skin drag), inelastic floe-floe collisions, breaking and rafting of floes, and many more," I personally believe it is unlikely that simple exponential attenuation via the linear differential equation da/dx =- $\alpha$ a will predominate. However, I also recognize that our ability to differentiate between it and a more sophisticated model such as da/dx =- $\alpha$ an may not be achievable because of the considerable uncertainty associated with most field data sets or remotely-collected data, e.g. satellite or airborne SAR, and that exponential decay may be perfectly reasonable for operational wave forecasting and large scale modelling projects. The authors could be a little more forthright in saying this, assuming that they agree with me.

**Yes, we do agree. This is exactly what we mean by saying "In many cases, large scatter in**

**observational data and/or limited number of measurement locations make the usage of more complicated models unjustified." (a sentence cited by the Reviewer above).**

3. In a similar vein, even if exponential attenuation is prevalent, it is very unlikely that a single attenuation coefficient  $\alpha$  will hold, given the intense heterogeneity associated with most fields of sea ice floes. The authors could add words to this effect too.

**Yes. The fact that $\alpha$ is variable and depends on both ice properties and wave characteristics seems so obvious that it might well be that we didn't explicitly wrote it. We will add a comment on that to the revised manuscript.**

4. While mostly saved for Part B, there is a statement on page 4 that even seemingly simple situations lead to wave propagation and attenuation that is shaped by several interrelated processes that are impossible to isolate from one another, and that several different model configurations can reproduce the observed attenuation rates. As the authors point out, this makes identification of processes actually responsible for dissipation a formidable task. This is a crucially important— although a rather dispiriting and potentially controversial conclusion, which needs to come through much more strongly in the paper(s) than it currently does, as it potentially redirects the topic to a more empirical/statistical future. Again, I encourage the authors to be more assertive about this conclusion as, while it is mentioned early in the paper and is reiterated in Part B, it appears to be played down or absent in § 5 of Part A.

The text on page 4 that the Reviewer refers to introduces the topics covered by both part A and B – it is a short introduction to the whole study. We state clearly that the issue of many different models/model configurations "explaining" the same dataset is discussed in part B, so that the reader should not expect to find it in part A. In our opinion, it would be confusing to elaborate on that in the discussion of part A, which is devoted solely to DEM modelling, model sensitivity, etc. The problem in question can be discussed only in the context of observational data, and therefore it would be strange to discuss it in part A, i.e., prior to the presentation of laboratory measurements. As for the conclusion itself: we don't find it "dispiriting", although it does "redirect the topic to a more empirical future" – in the sense that, apart from the attenuation rates alone, other measurements are necessary to distinguish between individual processes contributing to attenuation. Actually, if we agree that many available observational datasets are not sufficient to distinguish between exponential and non-exponential form of wave attenuation (and comment #2 makes us believe that we do agree on that point), it should not be surprising that several models can be proposed that "explain" those observations with similar accuracy.

In fact, how this problem is approached depends to a large degree on the purpose of a given model. If the goal is simply(!) to achieve good predictive skills in terms of the simulated wave attenuation rates, it might not be important to capture details of all physical processes involved. If, however, the goal is to understand the underlying physics, our case study shows that simultaneous measurements of several variables, and not the attenuation rates alone, are necessary to constrain the model parameters.

But, as said, this aspect of our results belongs to part B, not to part A.

5. As noted above, the DEM model is one-dimensional. Not much is made of this but it may have serious implications that the authors need to discuss. While Part A includes only the transmitted mode, Part B assimilates more modes using the Kohout et al. (2007) analysis. In both cases all the

**energy is trapped in the wave vector direction. What is the effect of this?**

Yes, as our DEM model was configured for the laboratory setup of the LS-WICE experiment, which was one-dimensional (as far as we can judge from the collected data, visual observation, video material, etc., it is justified to treat the waves in that experiment as unidirectional), all simulations are relevant only for this very special case. This is, obviously, a serious limitation (similarly as the fact that we consider only monochromatic waves), and it is not trivial to extend those results to a 2D situation with a full wave energy spectrum.

**It is a subject for subsequent studies, but we agree that we should comment on that in the discussion of the revised paper.**

6. One of the authors (Shen) has constructed an impressive, sophisticated model of sea ice to study wave propagation that includes a form of viscosity via a complex shear modulus. As such it is more general than the options provided by dispersion relation (6), which restricts us to open water, the mass-loading approximation and the elastic plate. There is also the simpler model of Robinson and Palmer (1990, doi: 10.1016/0022-460X(90)90661-I) that includes viscosity by means of a velocity-dependent damping term, and which seems to reproduce observations better in regard to the frequency dependence of  $\alpha(\omega)$ . Especially given that the mass-loading model tends not to be favoured by many of my colleagues, I ask the present authors to provide an explanation as to why they used dispersion relation (6). (Note that I am only asking for an explanation, not a reworking of the model.)

**There were at least two reasons that we decided to use dispersion relation (6).**

The practical one is that all variables it contains (elastic modulus, ice density, etc.) were known from measurements. The usage of another dispersion relation, dependent on the viscous parameter of the ice, would introduce a new unknown to our analysis (direct measurement of the viscous parameter has been shown in the past several decades as a very challenging task; the authors have attempted such measurements in the lab but did not succeed).

The second reason was consistency between the dispersion relation used and the assumptions underlying the DEM model (individual ice floes, elastic interactions between them). It is also worth pointing out that equation (6) was used in the previous analysis of the LS-WICE data by two of the present authors (Cheng et al., JGR, 2018). We do believe that equation (6) well represents LS-WICE observations (see Fig. 2 in part B), and that viscous damping in those experiments was not significant (notably, the ice floes floated in clear water, as opposed to many observations of wave damping in the MIZ, where the presence of frazil/pancake mixture gives the surface ocean layer high effective viscosity).

In our opinion, an important conclusion from our study is that the wave attenuation *is very sensitive to wave group velocity, and thus to dispersion relation.* We demonstrate it on the example of equation (6) and its two limiting cases (EP and ML) – but the conclusion is more general (we comment on that further in response to comment #9). We agree that we should add the above explanation to the revised manuscript.

7. The authors point out the difference between the DEM in regard to the attenuation of wave energy and the Shen and Squire (1998) model, and they include source terms for both drag and overwash, the latter primarily targeted at the experiments reported in Part B and being especially important near the ice edge before waves have attenuated substantially. I understand the explanation for the difference but what is the effect of choosing this particular formulation?

One of the effects that we discuss in our paper is the increase of attenuation rates with increasing restitution coefficient. The SS98 paper combined the ice and water systems as one. Hence they could attribute the energy decay of the combined system to floe collisions. The work done by the drag force becomes the internal exchange of the energy between ice and wave. In the present paper, the ice and water are considered as separate systems. In this way, damping due to drag force is explicit, as a positive source term in the ice energy equation, and a sink term in the water wave equation, simultaneously. This treatment is logically more rigorous.

In our present model, *E* represents the energy of the waves, which is affected by collisions only indirectly, through ice-water drag, so that there is no collision source term in the energy transport equation, and *E* is lower with higher  $\varepsilon$ .

8. I really enjoyed the c = 1 analysis, especially the outcome that da/dx =  $-\alpha^c a^2$ , with its solution a(x) =  $1/(\alpha x + 1/a_0)$ . This notwithstanding, I was bemused by why the limiting case of confined ice, i.e. with no collisions, should satisfy a particular exemplar of the more general differential equation predicted by Shen and Squire (1998) when collisions were modelled, viz. da/dx =  $-\alpha a^n$  with n = 2. I guess this reinforces point 4 above, paraphrased that different physics can lead to the same outcome. I also enjoyed the subsequent analysis that investigated how  $\alpha$  varied for the three different dispersion relations used, revealing  $\omega^4$  behaviour for open water,  $\omega^{>4}$  behaviour for the mass-loading approximation and  $\omega^{<4}$  for the continuous elastic plate. The asymptotic dependence of  $\alpha(\omega)$  is important because of the consistency of in situ field measurements that suggest a power between 2 and 3.

**Yes, this example illustrates that different models might produce the same, or very similar, a(x) and $\alpha(\omega)$ behavior.**

9. As hinted at earlier, I would contest the statement made at the end of § 3 that the mass-loading model is a good approximation of waves propagating through small ice floes or, at least, ask the authors to provide a reference to that effect. However, I do agree with the sentiment expressed that, for this model, it is the dispersion relation (and  $c_B$ ) that causes differences in how  $\alpha$  behaves. Because, taken out of context, this is counter-intuitive as one expects dispersion relation (6) to provide information about how the principal propagating mode disperses rather than attenuates, the authors could explain this better.

Regarding the second part of this comment: with the wave energy transport equation of the form  $d(c_g E)/dx=S$ , the change of wave energy with distance  $dE/dx=Sc_g^{-1}$  (assuming constant  $c_g$ ), i.e., the dispersion relation has a direct influence on wave attenuation through its influence on group velocity – independently of how the source term S is formulated. We point this out in the discussion section.

As for the choice of equation (6), it must be remembered that in the LS-WICE setup analyzed in our work (small ice thickness), the difference between open water (OW) and mass loading (ML) wave numbers is very small, so that our analysis of how different  $c_g$  affect  $\alpha$  could be made for OW versus EP (instead of ML versus EP) with very similar results. As we already mentioned in our reply to comment #6: the analysis by Cheng et al. (2018; see also Fig. 2 in part B) confirms that in LS-WICE the wavenumber increased with decreasing floe length, and the wavenumbers for the smallest floes considered ( $L_x = 0.5$  m) were slightly above those computed from the open water dispersion relation. The authors feel that mass loading is a reasonable model for small elastic floes as far as the dispersion is concerned. This is because at the free edges of each floe there is no bending hence no

storage of elastic energy even though the floes are elastic. In this way, the whole ice cover with these small floes practically approaches a collection of discrete rigid bodies. Once more, in the field, wave attenuation in small floes is usually analyzed for pancake/frazil mixture, and it is a very different ice type from that considered here. We agree that we should state that more clearly in the revised paper.

10. The paper continues by using the model derived in earlier sections with some of the laboratory data from Part B, essentially embarking on a sensitivity study to see which parameters influence specific outcomes. Some results are not obvious, e.g. why an increased restitution coefficient should lead to lower wave amplitudes, but this is explained. A change of slope is very evident in many circumstances in Fig. 2, dividing the ice field into two zones of high and low attenuation as is often observed in the field (see Squire and Moore, 1980, e.g.). Comparable curves in Fig. 2a and 2b show the effect of the mass-loading and elastic plate dispersion relations. The figure is well explained and the point is well made that smaller floes will always lead to stronger attenuation because of at least two mechanisms. Floe size is investigated in Fig. 5, again with the elastic plate dispersion relation showing a smaller reduction in a(x) than the massloading one, while average floe to floe distance is considered in Fig. 6a, where similar behaviour is evident save for very short waves where the zone of intense attenuation is particularly narrow. How the initial amplitude ao = a(0) affects a(x) is shown in Fig. 6b. In sum, the properties and the outputs of the model are well tested and well explained by a number of figures, of which I have mentioned just a few. The authors are congratulated for their thorough analysis.

**Thank you!**

11. The authors begin § 5 on page 16 by reminding us that a(x) and  $\alpha(\omega)$  are signatures of the underlying dissipative physics; the question is can the behaviour of these parameters be used to shed light on that physics? We are told that "the DEM simulations predict a very distinctive pattern of wave attenuation resulting from ice-water drag and collisions," but, as the authors point out, conditions at a real ice edge may be dominated by compressive forces exerted by radiation stress, winds and currents compensating increased granular pressure within the ice cover sustained by the waves themselves. Furthermore, the wavelengths considered in the simulations are comparable to the size of the floes, which is different from the situation that is often seen in the winter Antarctic and more recently in the western Arctic where a pancake ice zone forms at the ice margin. What is particularly noteworthy—and deserving more comment to finish—is the result that the tail of  $\alpha(\omega)$  follows a power law  $\omega^{2-2.5}$  within the ice interior, quite different from open water and close to what has been observed in situ.

**We write this in the abstract and in section 3, where this result is obtained, but we agree that the form of $\alpha(\omega)$ in the ice interior should be mentioned again in the discussion section. We will add that to the revised version.**

**Technical corrections**

I have very few technical corrections that I ask the authors to address, as follows

1. Page 3, line 5. Squire (2018) is not really a review paper. Rather, that paper justifies and then fleshes out an explanation for the differential equation  $da/dx = -\alpha a^n$  as a power-law fluid and tests the outcome of different values of n. The word review should be removed.

**We agree. The word "review" will be removed.**

2. Page 3, line 13. The Part B paper is not in the bibliography.

At the stage of submitting it was not clear how to cite B in A and vice versa. We will add the proper references to both bibliographies in the revised version.

---

## Author Comment (AC2) · 12 Sep 2019

**Response to the comments of Reviewer #2**

This paper deals with theoretical aspects relevant to the dissipation of waves traveling in sea ice. Particular focus is devoted to the role and parameterization of the ice-water drag as relevant dissipation mechanism. A discrete-element model (DEM) was employed in order to simulate the motion and collisions of the ice floes under the wave action, coupled to the wave energy transport with phaseaveraged source terms. As the aim of the paper was to explain wave dispersion and attenuation observed in a wave channel, wave energy dissipation due to overwash was also considered for completeness. Indeed, laboratory wave data are the subject of a companion paper (Part B), for which I was also asked to review. Unlike the water-ice drag, a minor role was recognized to the overwash mechanism to account for wave energy dissipation. Wave energy attenuation was analytically analyzed in the case of compact, horizontally confined ice cover. Interestingly, the authors show that a nonexponential wave attenuation law with the distance has to be expected if a quadratic drag law at the icewater interface is assumed. Current wave field observations do not allow to discriminate between the widely accepted/assumed exponential wave energy decline against other types of wave attenuation as a result of the large data scatter provided by in situ wave measurements. This means that new technologies should be envisaged to overcome this experimental limit. Authors also show that the attenuation rate is frequency-dependent and the dependence is related to the dispersion relation used. To this end, the authors assumed a wave dispersion relationship which blends shortening (mass loading) and lengthening (elasticity) of the open sea wavelength proportionately to the nature and rheology of sea ice. I support this paper. Some specific comments will be reported below, which I would like to read in the final version of the paper: 1) a discussion to explain the choice of the wave dispersion (eq. 6) could be added. The reason is the presence of the mass loading term. The weak point is that it could not adequately represent the ice floes assumed in the paper in terms of horizontal size/ wavelength ratio. In fact, the mass loading term is considered valid for really point-like ice floes (compared to the wavelength). 2) The relevance of papers like this is the possibility to extrapolate to the real world what learned for the in-door environment, also in simulation. So, to what extent do the authors think their model formulation can represent the complexity of our changing Arctic and Antarctic MIZ?

Thank you for the very positive reception of our paper and for all comments.

1) Speaking of "the mass loading term" is a bit misleading, as this term is present in Eq. (6) in its full form, in other words, both the elastic energy and the inertia effects are included in the thin-elastic plate theory.

Answering the doubts related to the usage of the dispersion relation (6), we will repeat our arguments from the reply to the comments of Reviewer #1:

"There were at least two reasons that we decided to use dispersion relation (6). The practical one is that all variables it contains (elastic modulus, ice density, etc.) were known from measurements. The usage of another dispersion relation, dependent on the viscous parameter of the ice, would introduce a new unknown to our analysis (direct measurement of the viscous parameter has been shown in the past several decades as a very challenging task; the authors have attempted such measurements in the lab but did not succeed).

The second reason was consistency between the dispersion relation used and the assumptions underlying the DEM model (individual ice floes, elastic interactions between them). It is also worth pointing out that equation (6) was used in the previous analysis of the LS-WICE data by two of the present authors (Cheng et al., JGR, 2018). We do believe that equation (6)

well represents LS-WICE observations (see Fig. 2 in part B), and that viscous damping in those experiments was not significant (notably, the ice floes floated in clear water, as opposed to many

observations of wave damping in the MIZ, where the presence of frazil/pancake mixture gives the surface ocean layer high effective viscosity).

In our opinion, an important conclusion from our study is that the wave attenuation *is very sensitive to wave group velocity, and thus to dispersion relation.* We demonstrate it on the example of equation (6) and its two limiting cases (EP and ML) – but the conclusion is more general (we comment on that further in response to comment #9).

We agree that we should add the above explanation to the revised manuscript."

2) Although our model and laboratory experiment are very simple (one-dimensional setup, monochromatic waves, etc.), we believe that several aspects of the results are practically relevant, e.g., the fact that collisions and overwash are likely to be relevant only in a narrow zone close to the ice edge, where they are responsible for very fast attenuation, and that turbulent dissipation due to ice-water drag is likely to dominate further downwave from the ice edge (as the most recent field observations from the Beaufort Sea seem to confirm; see Voermans et al. 2019).

Regarding part B of the study, the conclusion that several different mechanisms might produce similar attenuation rates, so that measuring wave attenuation alone is not sufficient to identify the underlying dissipative processes, is extremely important and practically relevant for both observations and modelling of wave energy attenuation in sea ice.